#### Development of the Horizontal Cloud Condensation Nuclei 1

#### Counter (HCCNC) to detect particle activation down to 4°C 2

#### temperature and 0.05% supersaturation. 3

- Mayur G. Sapkal, Michael Rösch and Zamin A. Kanji
- Department of Environmental System Sciences, ETH Zurich, 8092, Zürich, Switzerland
- 7 Correspondence to: Zamin A. Kanji (zamin.kanji@env.ethz.ch) and Mayur G. Sapkal
- (mayur.sapkal@env.ethz.ch)

Abstract. Aerosol particles play a critical role as cloud condensation nuclei (CCN) in the atmosphere. The capacity of aerosol particles to activate into cloud droplets is measured experimentally using CCN counters (CCNC). Recent findings suggest that the co-condensation effect of semi-volatiles can enhance aerosol particle growth and cloud droplet activation. Conventional CCNCs, such as the streamwise CCNC, heat particles as they transit the CCNC column and may inadvertently not capture the co-condensation effect leading to an underestimate in CCN concentrations. Additionally, streamwise CCNC struggle to achieve supersaturations below 0.13%, limiting their applicability for studying hydrophilic particles like (NH<sub>4</sub>)<sub>2</sub>SO<sub>4</sub> larger than 111 nm. To address these limitations, we developed the Horizontal Cloud Condensation Nuclei Counter (HCCNC), that can generate supersaturation at temperatures down to 4°C and supersaturations down to 0.05%. This study presents the development of the HCCNC, providing a detailed technical description of its 3D geometry, computational fluid dynamics simulations and the key components that demonstrate its performance, showing accurate performance at low temperatures and SS which the widely used commercially available Droplet Measurement Technologies Inc.(DMT) CCNC cannot achieve. The main chamber parts were 3D metal printed from an aluminum alloy. Sampling and humidity generation followed the principle of the previously used continuous flow thermal gradient diffusion chambers. Particles were detected using a commercially available optical particle counter (OPC, MetOne Instruments, Inc., Model 804). The instrument's performance is validated by conducting laboratory tests using ammonium sulphate ((NH<sub>4</sub>)<sub>2</sub>SO<sub>4</sub>) particles in the size range between 50 and 200 nm and for temperatures between 30°C and 8°C. Future work will focus on exploring the co-condensation effect on cloud droplet activation of levoglucosan and ammonium sulphate particles.

## Introduction

Aerosol particles serve as CCN for cloud droplet formation in the atmosphere. The key parameters determining aerosol CCN activity and ultimately cloud droplet formation under atmospheric water saturation include particle size and composition. Conventional Köhler theory can predict the development of cloud droplets by condensation of water vapour onto particles (Köhler, 1936), by combining the Kelvin effect and Raoult's law to estimate the saturation water vapor pressure above a curved aqueous solution surface. The capacity of aerosol particles to activate into cloud droplets is measured experimentally using CCNCs, which detect the proportion of particles that develop into cloud droplets when exposed to water vapor at a pre-set supersaturation (SS). Early laboratory research revealed that common soluble inorganic compounds of atmospheric aerosols, such as ammonium sulphate and NaCl, satisfy Köhler theory to a considerable degree, with measured critical supersaturations in agreement with theoretical predictions based on the Köhler theory such that closure is achieved (Cruz and Pandis, 1997). However, poor agreement is found between observed versus predicted CCN concentrations in closure studies (i.e., [CCN]<sub>observed</sub> ≠ [CCN]<sub>predicted</sub>) (L. Kammermann et al. 2010; Whitehead et al., 2014; Pajunoja et al., 2015). Potential explanations for these discrepancies arise from the uncertainty in aerosol composition due to internally mixed complex in/organic aerosol (Lee et al., 2002; Murphy et al., 2006), the corresponding surface tension deviating from that of pure water, in particular if surface active species form part of the aerosol (Ovadnevaite et al., 2017). Non-idealities that result from real world aerosol need to be accounted for to accurately predict CCN concentrations (Renbaum-Wolff et al., 2016; Ovadnevaite et al., 2017). Gas phase partitioning into hygroscopic aerosol increases its mass (and size) and in turn its propensity to take up more water (or activate at lower SS). This so-called co-condensation effect of gas-phase semi-volatile organic compounds condensing with water vapor is dependent on relative humidity and temperature of the aerosol and surrounding environment (Topping et al., 2013).

To estimate the critical supersaturation (SScrit) for particles to activate to cloud droplets, two types of CCNCs are utilized in laboratory studies: streamwise thermal-gradient diffusion chamber and parallel plate thermal-gradient diffusion chambers (CFDC). The streamwise CCNC designed by Roberts and Nenes (2005) and commercially available from Droplet Measurement Technologies, Inc., achieves water supersaturation (SS) exploits the faster water vapour diffusion of water compared to air by maintaining a temperature gradient within the chamber that is established using thermoelectric heaters. The chamber is made of a cylindrical column with wetted interior surfaces as a source of water vapor. The aerosol and sheath flow are drawn from the same source at the same temperature, while the sheath flow is passed through a filter followed by a Nafion membrane to achieve 100% relative humidity (RH), which keeps the column moist. At the top of the column, where the aerosol enters, the temperature is typically set to 0-10 K above the sample temperature to be above the dew point of the sampled air. This elevated temperature is essential to prevent condensation and fog formation, as the sheath air is at 100% RH (Roberts and Nenes, 2005). In environments with substantial air conditioning, if the ambient temperature falls below the dew point of the sampled air, condensation can occur, leading to aerosol changes. As heating progresses down the column, the temperature at the bottom is significantly higher than at the top. The temperature gradient, aligned with the aerosol stream, exposes particles to elevated temperatures throughout their transit, potentially altering their composition, particularly for volatile and semi-volatile species which can repartition into the gas phase (Hu et al., 2018). Additionally, the wet column walls act as a sink for water-soluble semi-volatiles, thereby removing these compounds from the activation process. As a result, these compounds do not contribute to cloud droplet formation under the experimental conditions, contrasting with their behavior in ambient environments (Romakkaniemi et al., 2014). Occurrence of this process has been estimated to induce considerable errors in the determination of the particle CCN ability (Hu et al., 2018; Topping and McFiggans, 2012). For instance, Hu et al., (2018) investigated the CCN behavior of (NH<sub>4</sub>)<sub>2</sub>SO<sub>4</sub> particles after exposure to propylene glycol (PG, PG = 0.1) and water (aw = 0.9). As the temperature difference in the CCNC column increased, the ratio of organics in the outlet was lower than that in the inlet, indicating the evaporation of organics. Consequently, they observed no significant changes in the kappa values or SS<sub>crit</sub> for (NH<sub>4</sub>)<sub>2</sub>SO<sub>4</sub> particles, whether measured with or without the presence of organic vapour. Additionally, the operational range of the streamwise CCNC is between 0.13% and 3% SS, with the lower limit constrained by the 1 µm droplet detection threshold, as the DMT CCNC utilizes this size as the criterion for identifying cloud droplets. Attempting to study the activation lower than the

supersaturation below 0.13% is challenging for two main reasons. According to Köhler theory, as the required critical supersaturation for activation decreases the dry particle diameter increases. Consequently, operating at lower supersaturation levels (than 0.13%) would necessitate a detection threshold greater than 1 µm to differentiate activated droplets from larger, unactivated particles. For instance, results from the E-AIM (IV) model indicate that an ammonium sulfate particle with a dry diameter of 191.4 nm would grow to approximately 1.4 µm at a relative humidity of 99.9% (saturation ratio, Sw = 0.99) (Wexler and Clegg, 2002; Clegg et al., 2013). This implies that the threshold size must be at least greater than 1.5 µm to classify the particle as a cloud droplet. Additionally, particles activated at lower supersaturations exhibit slower growth rates, as the supersaturation the primary driver of post-activation growth—is reduced compared to conditions above 0.13%. As a result, particles would require extended residence times to surpass the 1 µm detection threshold, with even longer residence times needed if the detection threshold were set higher than 1 µm. However, since the residence time in the streamwise CCNC is fixed for a given flow rate, operating below 0.13% supersaturation is impractical and does not allow for reliable cloud droplet detection (Roberts and Nenes, 2005; Rose et al., 2008; Tao et al., 2023; DMT CCNC manual). Moreover, according to the DMT CCNC manual, if particle concentrations exceed 6000 particles/cm<sup>3</sup>, the supersaturation lower limit considered as 0.2% to avoid counting biases, which arise from the growth kinetics of the particles. The streamwise CCNC therefore can present limitations to study atmospherically relevant particle sizes and different compositions (Rose et al., 2008; Tao et al., 2023; DMT CCNC manual). For example, as per the Köhler theory, the critical SS for 111nm ammonium sulphate particles is 0.13%, while for 84 nm particles it is 0.2%. As a result, studying particles larger than 111 nm in the streamwise CCNC may not be feasible or reliable. The presence of surfactants and the co-condensation of semi-volatile compounds can reduce the critical supersaturation required for activation at a given particle size, compared to conditions where these compounds and processes are absent (Nozière et al., 2014; Gérard et al., 2016; Topping & McFiggans, 2012). This suggests that a streamwise CCNC may be unable to detect the activation of such particles mixed with surfactants or processes like co-condensation occurring in the atmosphere.

CFDCs work by keeping two parallel plates at different temperatures, resulting in a linear temperature gradient between the plates. The inner surfaces of the plates are wetted such that the air near the walls becomes watersaturated, resulting in water SS (SS<sub>w</sub>) region between the plates (Kumar et al., 2003; Jiusto, 1967). In the CFDCs, aerosol flow is typically injected into the center between the two plates. If the temperature of both plates is adjusted simultaneously at a constant rate, the particles experience a consistent mean temperature throughout their transit. However, in earlier CCNC designs, this mean temperature was generally higher than the ambient temperature. Kumar et al. (2003) reported that the mean temperature of the chamber was maintained approximately 2°C above ambient temperature to mitigate evaporative losses of activated droplets before their detection by an Aerosol Particle Sizer (APS 3321, TSI Inc.). Additionally, the sheath flow was humidified to 100% RH by passing it through a fritted water bubbler before entering the chamber, which prolonged the wetting of the filter paper. However, humidifying the sheath flow poses limitations when attempting to achieve supersaturation (SS) at lower temperatures. Since the sheath flow is fully saturated at room or experimental temperature, cooling the plates below this level can lead to condensation of the sheath flow on the plates. This condensation will not only disrupt the established supersaturation profile by introducing excess vapor but may also result in the formation of droplets that could be sampled by the OPC. These unintended droplets may be mistakenly counted as activated particles, introducing potential counting biases. Collectively, these factors could compromise the reliability of critical

supersaturation measurements and impact the accuracy of activation determinations. So, far CFDC measurements are limited to supersaturations larger than 0.1%, given the long growth time required at lower SS, so droplets can be distinguished from interstitial aerosol (Nenes et al., 2001).

Dynamic spectrometers utilize a series of parallel warm and cold plates to create a variable supersaturation field, exposing aerosol samples to this field while inferring the CCN spectrum from the droplet size distribution measured at the outlet. Dynamic spectrometers have thus far reported measurable SS<sub>crit</sub> ranges from 0.01% to 1%. However, compared to CFDCs, dynamic spectrometers employ a fundamentally different design, in terms of particle activation and size measurement (Hudson, 1989). These instruments generate varying SS profiles through sequential heating in different sections and infer the CCN spectrum based on the droplet size distribution at the instrument outlet. A significant challenge arises when calibration curves derived from pure salt aerosols are used to infer ambient CCN spectra. Matching the calibration curve is often not satisfied for aerosols containing surfactants or slightly soluble materials, leading to uncertainties in measurement accuracy (Nenes et al., 2001).

In summary, several CCN instruments have been developed over the years, including the DMT streamwise CCNC, CFDC-based CCNCs such as described by Kumar et al. (2003), and the dynamic CCN spectrometer, amongst others, as reviewed in Nenes et al. (2001). Notably, none of these instruments can generate SS at lower temperatures (<10°C), which are not representative of natural cloud formation processes that occur at lower temperatures. Addressing these limitations is critical, as processes like organic repartitioning into the gas phase due to the higher temperature can significantly underestimate cloud droplet number concentrations compared to scenarios with co-condensation. According to Topping et al. (2013), the net cooling influence on global albedo, resulting from a 40%, 20%, and 10% enhancement in cloud droplet number concentration, is estimated to be 1.8 Wm<sup>-2</sup>, 0.98 Wm<sup>-2</sup>, and 0.51 Wm<sup>-2</sup>, respectively. While some instruments, like CFDCs, can operate slightly above ambient temperature, they often face difficulties in achieving SS below 0.1%. This limitation restricts their ability to measure CCN activity over the full range of particle sizes that are relevant for the atmosphere. The limitations of existing CCNCs to generate supersaturation over a wide range and at lower temperatures reveals a critical gap and underscores the aim of the work presented in this study.

Here we present the development of the Horizontal Cloud Condensation Nuclei Counter (HCCNC), a novel instrument designed to address challenges associated with low-temperature and low-supersaturation measurements. Although the HCCNC operates on the principles of a CFDC, it can generate SS over a wide range of sample temperatures (from 33°C - 4°C) that are more atmospherically relevant for cloud formation, while also achieving lower SS to sample a larger particle size range. A detailed technical description of the instrument is given and validation experiments to verify the accuracy of the achieved SS in the HCCNC are presented using ammonium sulphate aerosol particles. The results from the validation experiments are compared to existing literature and evaluated against Köhler Theory to verify accuracy and reliability.

### 2. Methods

# 2.1. Working principle

- The key components of the HCCNC are two temperature controlled parallel plates separated by a polymer spacer.
- The inner walls of the plates are lined with wetted filter paper and sandwich a polymer spacer (see Fig. 1).

Maintaining the two top (warmer) and bottom (cooler) plates at slightly different temperatures establishes a linear temperature gradient in the spacer region between the plates. Since the inner surfaces of both plates are wetted, a steady-state linear partial pressure of water develops between the cold and warm plates. As the equilibrium vapor pressure of water as a function of temperature is nonlinear, this results in a supersaturated environment with respect to water in the spacer cavity, with maximum supersaturation ( $SS_{max}$ ) occurring near the center of the cavity (See Fig. 2a). As a horizontally oriented CCNC, the temperature at the centerline is constant unlike the streamwise CCNC, where the lower section of the column has higher temperatures, which can drive buoyancy-induced air movement that opposes the intended downward flow and may create turbulence, particularly when the temperature difference within column exceeds 10 K. This turbulence not only disrupts the laminar flow but also displaces particles from the centerline and may equalize the temperature and water vapor profiles, potentially altering the set SS conditions (Rogers, 1988; Stetzer et al., 2008; Brunner and Kanji, 2021). In the HCCNC, the aerosol sample flow is sandwiched between a particle-free sheath flow, which is typically set to at least ten times that of the sample flow. The total flow within the cavity is maintained in the laminar regime to ensure that the aerosol sample remains centred between the particle-free sheath flows, thus experiencing uniform SS conditions. Since SS cannot be measured directly, the SS experienced by particles is inferred from temperature measurements and flow rates, with associated uncertainties detailed in Fig. 2b and discussed extensively in Section 3.2.

Figures A1, A2 and A3 present the results of computational fluid dynamics (CFD) simulations conducted using ANSYS software FLUENT (ANSYS Inc., 2010). In these simulations, the top plate was maintained at 8°C, the bottom plate at 5°C, while the sample and sheath flow, composed of dry nitrogen, entered the chamber at 20 °C. Figure A1a displays the formation of a linear temperature gradient between the two plates in the vertical direction, with the scale limited to 8 °C; temperatures exceeding this value were clipped. Figure A1b illustrates the midplane temperature profile, which shows slight disturbances as aerosols enter but stabilizes shortly thereafter. As a result of the temperature gradient, a SS profile develops, with the SS<sub>max</sub> approximately along the chamber centerline, as shown in Figure A2. While aerosol inflow initially at 20°C disturbs the SS profile, which quickly stabilizes. A detailed description of the CFD simulations is provided in Appendix A1.

### 2.2. Temperature Modulation in the HCCNC

Precise temperature control is a critical aspect of the functionality of CCNCs, particularly of the HCCNC as it directly influences the SS. The temperature modulation is facilitated by the top (warm) plate and the bottom (cold) plate (Fig. 1). The design of the HCCNC plates and associated components was developed using the CFD software, FLUENT (ANSYS Inc., 2010). After extensive simulations of various designs under different conditions, a plane-cavity-based design was chosen for its efficient coolant distribution and therefore uniform temperature across the plate (see section A2). The bottom plate differs slightly from the top plate in terms of design. The top plate features water-wetting ports, while the bottom plate includes a small drainage channel designed to collect excess water which can be removed via drainage ports (see next section). The cavity design posed notable challenges for conventional manufacturing methods, therefore 3D printing was utilized for fabrication. The plates were 3D printed from an aluminum alloy AlSi10Mg, selected for its compatibility with available 3D printing processes, as pure aluminum or other 3D metal printing options were not accessible during fabrication (ProtoShape GmbH). AlSi10Mg possess a low density of 2.67 g/cm³, good mechanical, thermal conductivity of 130-150 Wm⁻¹ K⁻¹, and chemical compatibility with coolants such as ethanol. The plates have the

dimensions of 410 mm (L) x 210 mm (W) x 13 mm (H). Compared to other parallel plate CCN and INP counters reported in the literature, the HCCNC plates are notably more compact, resulting in a reduced total heat capacity. This allows the usage of less expensive chillers with low heat load capacities while the temperature of the plates can still be increased or decreased rapidly which facilitates efficient changes in SS cycles. The plate temperatures are regulated using two external recirculating chillers (Lauda Proline RP855 and Lauda Eco RE620) (not shown in Fig. 1). Each plate is equipped with three thermocouples (Transmetra, TEMI313-K05-200-MS), inserted into the center of the plate beneath the cavity, which provides temperature readings at one-second intervals. The cooling fluid is distributed to the plates through a chiller manifold (see Fig. 1). As shown in Fig. A4, the internal diameter of the manifold gradually decreases along the streamwise direction, ensuring even distribution of the cooling fluid across all four inlets of the plates.

**Figure 1.** Exploded view of the horizontal cloud condensation nuclei counter (HCCNC), showing the entire chamber excluding the re-circulating chillers.

#### 2.3. Filter paper wetting procedure and water vapor source

The inner surfaces of both walls are lined with a single layer of self-adhering borosilicate glass microfibre filter paper (PALL 66217, 1  $\mu$ m, 8×1") (Fig. 1). The filter paper is wetted using a peristaltic pump operating at a flow rate of 10 ml min<sup>-1</sup>, delivering 40 ml of Milli-Q water from a reservoir to each of the three wetting ports located on the top plate before the experiment. Two of the three ports, supply water to the filter paper on the top plate, while the third supplies the bottom plate. To ensure complete wetting of the filter paper on both plates, approximately 150ml of Milli-Q water is pumped into the chamber and the excess (~100 ml) is drained through a peristaltic pump via a drainage port in the bottom plate. To facilitate uniform water distribution across the width of the plates, the setup is tilted by 25° parallel to the length axis of the HCCNC using a linear actuator. This procedure is performed once before initiating a new set of experiments. It has been observed that the wetted filter paper can consistently act as a water vapor source for up to 12–15 experiments consisting of a SS ramp from 0 to 0.8%, each consisting of 30-minute SS scan cycles with average ammonium sulphate aerosol concentrations from  $300-900 \text{ cm}^{-3}$ .

#### 2.4. Sample injection

The spacer made of polyvinylidene fluoride (PVDF) serves as a physical and thermal barrier between the two plates. The cavity is created by a cut-out within the spacer. The ogive shape of the spacer was chosen after evaluating multiple design options through simulations conducted under laboratory boundary conditions in ANSYS software (see section A1). The sample particles are introduced into the chamber through an injector via a slit measuring 75 x 0.40 mm. The injector is made of stainless steel, with an internal diameter of 4.57 mm and a length of 250 mm. The injector can be mounted in any of the eight available ports of the spacer (see Fig. 1), enabling precise control over particle residence time within the chamber and mitigating sedimentation issues that would arise from the horizontal orientation of the CCNC.

**Figure 2.** Supersaturation (SS) profile between the two parallel plates in the HCCNC (panel a). Example with cold wall at  $6.3 \,^{\circ}$ C (average of TC2 =  $6.07 \,^{\circ}$  and TC3 =  $7.41 \,^{\circ}$ C) and warm wall at  $8.2 \,^{\circ}$ C (average of TW2 =  $8.11 \,^{\circ}$  and TW3 =  $8.38 \,^{\circ}$ C),

shown by the purple line. The blue solid line represents equilibrium water vapor pressure, while the blue dotted line shows the steady-state partial pressure across the chamber. The black curve indicates the water SS profile. Schematic illustrating a side view of the HCCNC including uncertainty values in SS due to aerosol width and temperature measurements (panel b). The black curve represents the SS profile between the two plates as shown in (a). The sample flow from the injector, with a dilution factor of 17, is shown as the shaded grey region. The dotted line indicates the increased sample width when the dilution factor is reduced to 11. TW and TC denote thermocouples installed on the warm and cold wall, respectively. Position (4) denotes the SS uncertainty in the central lamina, resulting from a temperature uncertainty of ±0.47 °C. Corresponding uncertainties at lower SS are plotted in Figure 4c.

#### 2.5. Flow control

The total flow rate of 1.5 L min<sup>-1</sup> through the HCCNC is apportioned into a sheath-to-sample flow ratio of 10:1. This ratio is maintained to avoid wall losses and to confine the sample flow to a thin layer minimizing the variation in SS and temperature that the aerosol is exposed to in the center of the chamber. The sheath flow is controlled using a mass flow controller (MFC, ALICAT, MC-20SLPM) and is introduced through four ports directed onto aluminum heat sinks on the bottom wall (see Fig. 1), allowing rapid cooling of the flow to equilibrate with the chamber temperature. The sheath flow then passes through a mesh that stabilizes the flow and minimizes turbulence, as described by Brunner & Kanji (2021). Figure A3 presents the turbulence kinetic energy, which is minimal, confirming laminar flow conditions within the chamber. The plain-weave type-304 stainless steel wire mesh, with a mesh size of 250 mesh per inch and a wire diameter of 0.04 mm, spans the full width and height of the spacer cavity. After the mesh, the combined stream of sheath and aerosol, flows horizontally in a laminar fashion through the chamber, where the SS and temperature profile forms according to Fig 2a. Particles with a SS<sub>crit</sub> lower than the SS<sub>centreline</sub> activate and grow into droplets. A cone shaped exit of the chamber focuses the sample and sheath flows, directing the airstream into an optical particle counter (OPC, MetOne Instruments, Inc., Model 804). The cone is angled to ensure that the total flow merges smoothly, avoiding turbulence. A pump (KNF, N8134ANE) downstream of the OPC maintains the total flow of the instrument which is also controlled by an MFC (ALICAT, MC-20S L min<sup>-1</sup>).

#### 2.6. Droplet detection

The OPC used to detect droplets at the outlet of the HCCNC is equipped with a 780 nm, 35 mW photodiode laser. The onboard electronics processor from MetOne categorizes and sizes detected particles into four selectable size bins (from >0.3 to >10  $\mu$ m). Although the OPC is calibrated for a flow rate of 2.83 L min<sup>-1</sup> as specified by the manufacturer, we operated it at a reduced flow rate of 1.5 L min<sup>-1</sup>. This adjustment was made to ensure adequate residence time for particles within the HCCNC, as higher flow rates reduce the time particles remain under SS conditions, impacting their growth. Intercomparisons between two OPC units of the same model, operated at different flow rates, revealed a 14% lower particle number concentration when the OPC was operated at 1.5 L min<sup>-1</sup> compared to 2.83 L min<sup>-1</sup> (See Fig. A5). This observed discrepancy is accounted for when operating the OPC at 1.5 L min<sup>-1</sup>. To minimize the exposure of activated droplets to external environmental conditions and to ensure their rapid entry into the scattering volume, the distance between the HCCNC outlet and OPC inlet is minimized (17mm). The residence time in this section is also reduced due to an increase in local flow velocity due to the cone shaped exit.

#### 2.7. Electronic interface

The electronic interface is important to the operation of the HCCNC, designed to control its various features and processes. The primary components of this interface include National Instruments (NI) modules, microcontrollers (Arduino), MFCs, actuators, and relays. These components are housed within a waterproof plastic enclosure for protection. Temperature measurements are collected through three analog-to-digital (A/D) input channels of the NI-9213 module, which offers a 24-bit resolution. The Arduino microcontroller manages the operation of water pumps and actuators facilitating the tilting mechanism of the HCCNC during the wetting process. The chillers and SS cycles are controlled through a self-programmed interface in Python 3.12. In contrast, OPC data is collected using COMET software provided by the manufacturer.

Figure 3. External components and flow setup for the validation experiment using ammonium sulphate aerosol.  $T_c$  and  $T_w$  represent the re-circulating chiller used to control the temperatures of the cold and warm plates, respectively. The blue and red arrows indicate the direction of the outgoing and incoming fluid to the chillers

### 2.8. Experimental setup - validation experiments

Figure. 3 shows the external components of the HCCNC and the flow setup used for the validation experiments. In the current configuration, the temperature of the top plate is regulated by a Lauda Proline RP855 chiller, while the bottom plate temperature is controlled by a Lauda Eco RE620 chiller. Ethanol is used as the coolant fluid in both chillers. The flow of the chilled fluid into the plates and the subsequent outflow of the warmed fluid are indicated by blue and red arrows in Figure 3. Three peristaltic pumps (only one shown for clarity in Fig. 3) wet the filter paper before the start of an experiment. Dry, particle-free nitrogen serves as sheath flow for the HCCNC, controlled by MFC1. An ammonium sulphate solution  $(0.1 \text{ g L}^{-1})$  is aerosolized using a constant output atomizer (model 3076). The wet aerosol then passes through a diffusion dryer (filled with a 4 Å molecular sieve) followed by a differential mobility analyzer (DMA, Model 3082, TSI Inc.) for size selection. The aerosol flow is split by a Y-connector to a condensation particle counter (CPC, Model 3072, TSI Inc.) and the HCCNC. The OPC was configured in this experiment to measure particles concentrations in the >0.5, >0.7, >1.0, and >2.5  $\mu$ m size bins. Initially, the flow to the HCCNC is directed through a high-efficiency particulate air (HEPA, Model 12144, Pall Inc.) filter for background measurements. Once a low background was confirmed, the valve was switched to sample the aerosol particles. The injector was placed at position 8 relative to the OPC to ensure the maximum residence time of the particles within the chamber (21 seconds) (for more details see section 3.1.2). All unused

positions were securely closed with plugs to seal the chamber. MFC2 downstream of the OPC is set to 1.5 Std L min<sup>-1</sup> and the sample air flow rate of 0.257 Std L min<sup>-1</sup> is determined by the difference between the flow exiting the spacer through the OPC and the sheath air entering the spacer (~10:1 ratio).

### 3. Results and discussion

#### 3.1. Validation of the HCCNC

Ammonium sulphate was used as the test species for evaluating the performance of the HCCNC due to its well-characterized hygroscopicity and its extensively studied CCN activation properties (Brechtel and Kreidenweis, 2000). Aqueous ammonium sulphate solutions were atomized, dried, and size-selected using a DMA with a sheath flow rate of 12.5 L min<sup>-1</sup>, while the HCCNC aerosol stream flow was maintained at 0.12 L min<sup>-1</sup> for the validation experiments.

#### 3.1.1. Mobility diameter and volume equivalent diameter

Validation experiments were conducted using mobility diameters of 50, 60, 100, and 200 nm. As the DMA classifies charged particles based on their electrical mobility (Knutson and Whitby, 1975), the mobility diameter for spherical particles is well-defined and corresponds to the particle's volume-equivalent diameter. However, since the mobility diameter is a measure of particle behavior rather than an intrinsic property, the relationship between particle shape and the electric field in the DMA must be considered for aspherical particles. ammonium sulphate particles exhibit asphericity (Zelenyuk and Imre, 2007), thus necessitating the conversion of mobility diameter to volume-equivalent diameter (Zelenyuk et al., 2006). To calculate the volume-equivalent diameter, the shape factors from Kaaden et al., (2009) were used. The conversion, described by Equation (1), neglects the Cunningham correction, as its influence becomes significant only at the third decimal place.

$$d_{ve} = \frac{d_m}{\chi}$$

In this equation,  $d_{ve}$  is the volume-equivalent diameter,  $d_m$  is the mobility diameter, and  $\chi$  represents the shape factor. A shape factor of 1.03 was applied for mobility diameters of 50, 60, and 100 nm, while a factor of 1.045 was used for 200 nm (Zelenyuk et al., 2006). After conversion, the mobility diameters of 50, 60, 100, and 200 nm corresponded to volume-equivalent diameters of 48.5, 58.3, 97.1, and 191.4 nm, respectively. All results presented in this study are based on volume-equivalent diameters. In this study, we refer to dry particle sizes < 100 nm as small particles, and those > 100 nm as larger particles.

#### 3.1.2. Particle activation, growth and sedimentation

Once particles experience SS exceeding the  $SS_{crit}$ , they activate and grow. Particles must grow to a size > 1  $\mu$ m to be classified as activated droplets, as hygroscopic growth alone cannot achieve such sizes for our selected dry particle sizes. This growth can be modelled using diffusional growth calculations, as described by Rogers (1988) and detailed in section A4. Figure A6 (a and b) illustrates the theoretical growth trajectory of a 97.1 and 191.4 nm particle after activation at a fixed temperature (30 °C) at different SS conditions. The plot also shows the particle residence time in the HCCNC relative to different injector positions (ix, x=1-8). According to the Rogers model,

even at the shortest residence time ( $\tau = 7$  sec) in the HCCNC (i1, closest to the OPC), a particle should grow to >2 µm at a SS of 0.1%. However, our experiential results indicate that these theoretical diffusional growth calculations overestimate the final particle size upon exiting the chamber, as the model assumes the target constant SS is experienced as soon as the particle enters the chamber and throughout the particle transit. In practice, the total flow takes some time to equilibrate to the set chamber conditions, exposing the particles to lower SS during the initial seconds. This effect is also evident in the simulation results, where the SS profile is disturbed when the sample and sheath flow were introduced at 20°C and the warm and cold plates are maintained at 8 °C and 5 °C, respectively. However, the temperature and SS profile stabilizes after a short distance, as shown in Figure A1 and A2. In our chamber, particles will settle out of the flow if they fall 9.5 mm from the centerline and hit the bottom plate. For instance, as illustrated in Figure A7, at injector position 8, which corresponds to a residence time of 21 seconds, a particle with a dry diameter of 97.1 nm, when activated in a supersaturation environment of 0.15%, could grow to a diameter of up to 6 µm according to Rogers' model. As droplet size increases following activation, so does their fall speed, with droplets of this size (6 µm) expected to sediment onto the bottom plate at approximately 20 seconds. This suggests that, theoretically, these particles would not reach the OPC. However, we detect particles and thus we believe due to the time required for the total flow to equilibrate with the chamber conditions, there is a delay in particle activation and thus growth. This delay allows particles to be detected by the optical particle counter (OPC) before they settle out of the flow. During the validation experiments, droplets > 1 μm were classified as cloud droplets for dry particles up to 100 nm in diameter, while droplets exceeding 2.5 μm were classified as cloud droplets for dry particles of 191.4 nm in diameter (see section 3.1.2). This approach was adopted for two primary reasons: first, activation was not clearly detectable in the 1 µm OPC channel as the droplets already appear to have grown to 1 µm prior to their activation. This observation is consistent with predictions from Köhler theory and results from the E-AIM (IV) model, which suggest that an ammonium sulfate particle with a dry diameter of 191.4 nm grows to approximately 1.4 µm at a relative humidity of 99.9% (saturation ratio, S<sub>w</sub> = 0.99) (Wexler and Clegg, 2002; Clegg et al., 2013). Second, the OPC used in this study has a 2.5 µm channel as the next available size above 1 µm.

### 3.1.3. Supersaturation Scans and CCN Activation in HCCNC at 30 and 20 °C

366367

All validation experiments were performed by stepping the SS (also known as S-scans) within the chamber. Initially, both plates of the HCCNC were held at the same temperature, corresponding to a SS<sub>w</sub> of 0%. Subsequently, the temperature of the bottom plate was lowered at a rate of 0.2 °C min<sup>-1</sup>, causing a gradual increase in SS within the chamber. Although the setup would allow both plates' temperatures to be adjusted, controlling the temperature of only the bottom plate provided finer control over SS generation (e.g., from 0.05 to 0.1% SS, the rate of change SS was 0.013 ± 0.008 per minute). For the performance evaluation of the HCCNC, S-scans were conducted at nominal temperatures of 30°C, 20°C, and 8°C for each particle diameter (48.5, 58.3, 97.1, and 191.4 nm). A minimum of three experiments were performed for each size and temperature, to ensure the reproducibility of the results. Typical sets of data for the activation of ammonium sulphate particles are shown in Fig. 4a and 4b for particle diameter, 48.5, 58.3, 97.1 and 191.1 nm at 30 and 20°C. A CPC was used in parallel to measure particle concentrations and to determine the activated fraction (AF), defined as the ratio of CCN (droplets measured in the OPC) to condensation nuclei (CN, dry particle concentration in the CPC). The AF values range from 0 (no activation) to 1 (complete activation), with an AF of 0.5 indicating the SS<sub>crit</sub> for CCN activation. Particle concentrations were maintained between 300-900 cm<sup>-3</sup> during the experiments.

Figure 4. The activated fraction (AF) plotted as a function of supersaturation (SS, %) for ammonium sulphate particles with dry diameter of 48.5, 58.3, 97.1 nm and 191.4 nm. The critical supersaturation  $\pm$  uncertainty is indicated in the legend for each curve shown and the dotted curve represents a double sigmoid fitted to the measurements. The different panels show SS scans performed at different temperatures indicated. Horizontal error bars represent the estimated uncertainties in SS, derived from a combination of temperature measurements and

aerosol layer thickness. Panels a and b have very small uncertainties which do not show up on the data, but are provided for the SS<sub>crit</sub> in the legend.

For all particle diameters, there is a clear transition from no activation at lower SS to a plateau region indicative of full activation. Although theoretically the AF should reach 100%, in practice it is typically observed between 70% and 90% during our experiments. Discrepancies of this magnitude are common in the literature for TGDCs (Cruz and Pandis, 1997; Corrigan and Novakov, 1999; Kumar et al., 2003). This can arise from various factors, such as the combined ±15% uncertainty (see section 3.2) from the OPC and CPC counting and the variation in SS experienced in the aerosol layer. If the aerosol layer is thicker than calculated, a small fraction of particles will be exposed to lower SS than calculated (see next section for details). The S-shape of the activation curve and the plateau at higher SS exhibit typical CCN activation behavior and illustrate the instrument's ability to detect and measure droplet activation and growth at low temperatures and supersaturations. A first plateau in the AF indicates the behavior of doubly charged particles (Lance et al., 2006) (Fig. 4). For example, in experiments involving 97.1 nm particles at 20°C, activation is observed beginning with a SS of 0.025%, plateauing at 0.12%. The half-rise (critical supersaturation), of the sigmoidal activation curve was determined to be at 0.07%. This value corresponds to a particle size of 158 nm, which is characteristic of the activation of doubly charged particles. The measured SS<sub>crit</sub> from the HCCNC for ammonium sulphate particles with dry diameters of 48.5, 58.3, 97.1, and 191.4 nm were 0.48, 0.37, 0.18, and 0.05%, respectively, at 30 °C. These values align well with predictions from Köhler theory (Köhler, 1936), which predicts SS<sub>Köhler</sub> = 0.47, 0.36, 0.17 and 0.05%, respectively at 30°C (For further details, see Section 3.4). Similarly, at 20 °C, the measured SScrit values of 0.54, 0.41, 0.20, and 0.06% closely match the theoretical SS<sub>Köhler</sub> values of 0.52, 0.39, 0.18, and 0.06%, respectively. The horizontal error bars in Fig. 4 represent the estimated uncertainties in SS, derived from temperature measurements and aerosol layer thickness as detailed in the next section.

### 3.2. Uncertainties in Supersaturation and Activated Fraction

The AF is governed by critical parameters such as particle count and the flow ratio between sheath and sample flow since the dilution controls the concentrations of CN in the chamber. Since SS cannot be measured directly, its accuracy within the HCCNC is inferred from temperature measurements, with the wall temperatures defining the SS field. Temperature readings are obtained from three thermocouples mounted on each HCCNC plate, ensuring precise determination of the temperature gradient across the chamber and, by extension, the SS profile (Fig. 2a and 2b). While three thermocouples are installed on each plate, only the two closest to the injector (TW2/TC2) and the chamber outlet (TW3/TC3) are used to calculate the mean wall temperature, as particles are exposed to cavity conditions only after the injector. From the experimental data, we determined that the thermocouples have an uncertainty of ±0.07 °C within the 30 °C – 20 °C range. This uncertainty increases to ±0.27 °C at 8 °C and ±0.38 °C at 6 °C. In Fig. 2b, TW1/2/3 and TC1/2/3 show the temperature readings of the warm and cold plates, respectively, from one of the validation experiments.

In these experiments, the sample flow (depicted as grey shading) enters the cavity through the injector, with a dilution factor of 17 (sheath: sample = 16:1, 16+1=17). The particles are exposed to a SS profile (thick black curve) representing the values of the S-curve from Fig. 2a, where the SS<sub>max</sub> is 0.203%. However, due to the uncertainties in the warm and cold plate thermocouples,  $\pm 0.27$  and  $\pm 0.38$  °C respectively, a combined uncertainty

of  $\pm 0.47$  °C =  $(\pm \sqrt{0.27^2 + 0.38^2})$  is introduced at the center of the chamber (Fig. 3, point 4). This temperature uncertainty translates into a SS uncertainty ±0.093 % that particles may experience (i.e., SScenter ±0.093 %) (Fig. 2b, point 2). Also, as detailed in Section 2.1, the sample flow is sandwiched between the sheath flow. Originating from a narrow slit, the aerosol particles are exposed to a relatively consistent and narrow range of SS. However, altering the flow ratio causes the aerosol stream to broaden or narrow vertically, thereby subjecting particles to a wider or narrower SS. This effect is illustrated in Fig. 2b, where a reduction in the dilution factor from 17 to 11 (dotted line) leads to a shift in the SS experienced by the outermost aerosol particles, from 0.203 ±0.001 % (point 4) to 0.203 ±0.002 % (point 3) which could lead to a lower AF. The largest source of SS uncertainty, whether from temperature or aerosol stream width, has been accounted for and is incorporated into the results presented in Figure 4. As shown in Figure 2b, the uncertainty arising from temperature is greater than that associated with aerosol stream width. Therefore, temperature-induced SS uncertainty has been used to represent the SS variability experienced by particles (see Fig. 4). It is also evident that the error bars in Figure 4c, corresponding to experiments conducted at 8 °C, are more pronounced compared to those at 20 °C and 30 °C (Figures 4a and 4b). This trend is attributed to reduced accuracy of the thermocouples used in this study at lower temperatures. This limitation and the necessary improvements have been discussed in Section 3.4. The OPC and CPC used in the validation experiments have a counting uncertainty of  $\pm 10\%$  each, resulting in a relative uncertainty of  $\sqrt{(10^2+10^2)}$ 

# $434 = \pm 15\%$ (rounded up) in the reported AF.

### 3.3. Particle Activation at Lower Temperature and Lower Supersaturation

Figure 4(c) presents the activation curves for 97.1 nm and 191.4 nm ammonium sulphate particles at 8 °C. The sharp activation curves validate the ability of the HCCNC to generate a supersaturation at lower temperatures. Most conventional CCNC designs, such as the DMT CCNC and the CCNC from Kumar et al. (2003), employ a humidified sheath flow to maintain a moist filter paper. This approach requires heating the column or plates to avoid fog formation upon aerosol introduction or to prevent condensation within the chamber. The HCCNC addresses these limitations by employing a dry sheath flow, enabling experiments to be conducted at lower temperatures while still avoiding condensation and spurious supersaturation generation when humidified sheath flow enters the chamber at lower temperatures. Furthermore, the peristaltic pump system efficiently wets the filter paper on-demand, eliminating the need for continuous humidification and allowing broader operational temperature ranges. The measured SS<sub>crit</sub> from the HCCNC for ammonium sulfate particles with dry diameters of 48.5, 58.3, 97.1, and 191.4 nm were 058, 0.46, 0.20, and 0.1%, respectively, at 8 °C. These measurements align well with the predictions from Köhler theory - 0.57, 0.43, 0.17 and 0.07%, respectively, given the experimental uncertainties.

Besides the low temperature operation, another novelty of the HCCNC is its ability to operate at very low supersaturation levels. The DMT CCNC and old CFDC's typically operate above 0.13 and 0.1 % SS respectively (Roberts and Nenes, 2005; Nenes et al., 2001). Figure 4 demonstrates the activation curve for 191.4 nm particles, which necessitate very low SS for activation (0.05%). The HCCNC precisely maintains the required temperature gradient to achieve these low SS conditions. For instance, to achieve a SS of 0.05% at warmer temperatures, a temperature difference of just 1.2°C between the two plates is required. The HCCNC's ability to operate at lower temperatures and to achieve low SS levels is largely attributed to its cavity-based plate and spacer design. The

metal thickness separating the filter paper from the cooling fluid is only 3 mm, which ensures precise temperature control and, consequently, accurate regulation of supersaturation. Also, the steep activation observed in the particles indicates that most of the particles were exposed to a consistent supersaturation, suggesting minimal turbulence within the chamber. This is supported by the calculated Reynolds number for the HCCNC, derived from turbulent kinetic energy (TKE) (Fig. A3) values obtained from CFD simulation results. The Reynolds number ranges from 0.009 to 395 in HCCNC, well below the turbulent flow regime threshold of 4000.

### 3.4. Validation of Köhler Theory and Non-Ideal Behaviour of Ammonium Sulphate

To validate the HCCNC, CCN activation experiments were conducted in the temperature range from 30°C - 8 °C to measure the SS<sub>crit</sub> for size selected ammonium sulphate particles. To verify accuracy, the SS<sub>crit</sub> measured was compared to the predicted SS<sub>crit</sub> calculated with standard Köhler theory and data from Davies et al. (2019). The SS values used for comparison (SS<sub>Köhler</sub>) were calculated using Eq. (2) as discussed in Kumar et al., (2003).

$$S_C = (4A^3/27B)^{1/2}$$
 2

where  $S_C$  is the critical supersaturation. The values of A and B calculated using Eq. (3) and (4):

$$A = (2\sigma_{sol}M_w)/(\rho_{sol}RT)$$

and

$$A71 B = (3vm_s M_w)/(4\pi \rho_{sol} M_s) 4$$

Where  $\sigma_{sol}$  is the surface tension of the solution and it is set to that of water for temperatures of 30 °C, 20 °C, and 8 °C, with values taken from Vargaftik et al., (1983) which are 0.071, 0.073 and 0.075 Nm<sup>-1</sup> respectively.  $M_w$  is the molecular weight of water (18.02 g mol<sup>-1</sup>), the density of the solution (water)  $\rho_{sol}$  is taken as 997 kg m<sup>-3</sup> and R is the ideal gas constant. The temperature (T) used is  $T_{activation}$  (i.e., temperature at which particle activation is observed during experiments) was 29.10, 18.90 and 7.20 °C corresponding to experiments conducted at 30, 20, and 8 °C respectively. The effective van't Hoff factor, v is 2.04 (Wu et al., 2011) and 1.94 (Rose et al., 2008; Frank et al., 2007, details follow in the next paragraph) for non-ideal behavior or 3 for complete dissociation,  $m_s$  is the mass of ammonium sulphate in the dry particle based on its radius,  $M_s$  is the molecular weight of ammonium sulphate (132.1 g mol<sup>-1</sup>).

Figure 5(a) shows a systematic discrepancy between the calculated and experimentally measured  $SS_{crit}$  ( $SS_{measured}$ ) using both HCCNC and DMT CCNC values, with the measured  $SS_{crit}$  generally being higher than the theoretical predictions. This difference may stem from uncertainties previously discussed arising from the temperature, SS variation in the aerosol layer. Additionally, the potential non-ideal behaviour of ammonium sulphate may contribute to the discrepancy. To explore this further,  $SS_{crit}$  was calculated using three effective van't Hoff factors (v) (Figures 5a - 5c) assuming: complete dissociation (v = 3), accounting for non-ideal behaviour (v = 2.04) recommended by Wu et al. (2011) and v = 1.94 calculated using the parameterisation in Rose et al. (2008, equation A25) and Frank et al. (2007). When non-ideal behaviour is accounted for, the agreement between measured and theoretical  $SS_{crit}$  improves significantly. We note that that Figure 5 also validates that Köhler theory requires a higher  $SS_{crit}$  for same sized particle at lower temperature (Eqns. 2, 3 and 4) and this outcome underscores the capability of the HCCNC to perform highly controlled CCN measurements under lower temperature conditions,

providing valuable insights that to the best of our knowledge have not been shown in any previous studies. Furthermore, the kappa ( $\kappa$ ) value derived from HCCNC measurements at 8 °C is 0.42±0.37. This result is comparable to the  $\kappa$  of 0.49 measured at 10 °C by Cheng and Kuwata (2023) a Low-Temperature Hygroscopicity Tandem Differential Mobility Analyzer (Low-T HTDMA). Similarly, HCCNC-derived  $\kappa$  at 20 °C is 0.47±0.03, which agrees with the value of 0.46±0.01 reported by Gysel et al. (2002) at the same temperature. The larger uncertainty in  $\kappa$  at 8 °C is attributed to the larger SS uncertainty at lower temperatures, as discussed in Section 3.4.

**Figure 5.** Comparison between experimental critical supersaturation (SS<sub>crit</sub>) values and theoretical SS<sub>crit</sub> (SS<sub>Köhler</sub>) predictions for ammonium sulphate, calculated using standard Köhler theory as outlined in Eq. (2), (3), and (4). Each symbol represents a fixed-diameter experiment, with the colour coding (red, green, and blue) indicating measurements performed at the depicted temperatures, also including results from literature. (a) shows the Köhler theory calculation assuming full dissociation ( $\nu = 3$ ), while (b) and (c) accounts for non-ideality with an effective van't Hoff factor of  $\nu = 2.04$  (Wu et al., 2011) and  $\nu = 1.94$  (Rose et al., 2008; Frank et al., 2007).

### 3.5. Applications and Limitations

Unlike conventional CCN instruments, which generate SS at higher temperatures, the HCCNC can achieve SS at lower temperatures, which is more representative of atmospheric conditions. This capability enables new opportunities to investigate phenomena such as co-condensation of organic vapours that would partition out of the particle phase at higher temperatures for temperature-dependent CCN analysis of ambient aerosols. Additionally, the HCCNC's ability to operate at very low SS levels enables the analysis of larger particles that are as hygroscopic as ammonium sulphate (requiring low SS<sub>crit</sub>) such as in the marine environment or of internally mixed in/organic compounds that have a lower hygroscopicity than pure ammonium sulphate but are larger than 200 nm. This extends the experimental range possible compared to the DMT CCNC, which is only reliable down to a supersaturation of 0.13%. This feature also facilitates investigations into surface tension reduction of the droplet, where the required critical supersaturation can be exceedingly low. In HCCNC, the largest OPC channel utilized for validation experiments to quantify droplets is 2.5 µm. To estimate the maximum particle concentration at which we can operate before water vapor competition becomes significant for 2.5 µm droplets, we calculate a particle concentration of 7400 cm<sup>-3</sup>, assuming a temperature of 30°C and SS of 0.2%. This value is subject to variation with changes in SS and temperature, with further details provided in Section A5, which accompanies Fig. A8. The HCCNC is lightweight and compact, indicating its potential suitability for future field campaigns.

However, two high-capacity chillers, which are both heavy and costly, were used primarily due to availability rather than necessity. While effective, these chillers are not essential for CCN measurements, as more affordable and lightweight options with lower cooling capacities could be sufficient for this application. This suggests that future iterations of the HCCNC could leverage alternative, more practical cooling solutions without compromising performance, thus enhancing the instrument's portability and cost-effectiveness for field deployments. The sample and sheath flow system in the HCCNC is simpler compared to vertical geometry setups. The plates were fabricated using 3D metal printing, a technology that is becoming increasingly widespread and accessible, suggesting that the cost is likely to decrease in the future. Another advantage of the HCCNC is its ability to operate with dry sheath flow, eliminating the need for nafion humidifiers and allowing operation at cooler temperatures. This not only reduces the complexity of the device/system but also minimizes the number of components, leading to lower maintenance requirements, reduced downtime, and enhanced overall user-friendliness. The wetting of the filter paper is managed on-demand by pumps, simplifying the setup and making it more adaptable for field use. Additionally, an OPC with more size bins or different size resolution can be deployed as well if longer droplet growth times or different flow rates are desired.

#### 4. Conclusions and outlook

In this study we have shown the development of the Horizontal Cloud Condensation Nuclei Counter (HCCNC), an in-house built parallel plate cloud condensation nuclei (CCN) counter designed to study CCN activity at low temperatures and supersaturations, while conventional CCN instruments typically operate under warmer conditions. The HCCNC follows the general operating principles of other CFDC instruments and was designed based on CFD simulations to achieve a better temperature uniformity and flow laminarity. The HCCNC operates with a dry sheath flow, enabling it to function across a wide range of temperatures without issues of condensation. The chamber is compact, lightweight and performs well over a wide range of temperatures when verified against SS<sub>crit</sub> for ammonium sulphate particles. Furthermore, its precise temperature control allows it to generate and operate at very low SS levels. Multiple injector positions provide flexibility in selecting particle residence time, which helps mitigate sedimentation issues that could be encountered for experiments conducted at higher SS<sub>crit</sub> in the future, for example with less hygroscopic species. To validate the performance of the HCCNC, monodisperse ammonium sulphate particles were used. The resulting activation curves from the supersaturation scan experiments were within experimental uncertainties, aligning with Köhler theory, and demonstrated reproducibility. Besides, this study also provides support for Köhler theory, which predicts that the critical supersaturation SS<sub>crit</sub> required for activation increases for particles of the same size as temperature decreases.

These features and technological advancements in the HCCNC have been filed as a patent. Commercialization efforts are already underway to make this instrument widely available, enabling the atmospheric research community to fully leverage its potential. Future experiments with the HCCNC will be conducted on organic/inorganic mixtures where the HCCNC will be operated in parallel with a streamwise CCNC during a measurement campaign sampling ammonium sulphate and levoglucosan mixtures at varying wet particle diameter assessing impacts of co-condensation on CCN activation. Further development of the HCCNC is with a focus on extending particle residence time to enable operation at even lower supersaturation levels is planned. Additionally, the thermocouples employed in this study exhibited reduced accuracy at lower temperatures, which contributed to higher uncertainty in supersaturation measurements. Future iterations of the instrument will address this

limitation by incorporating high-precision thermocouples to improve temperature control and measurement accuracy. Lastly, future upgrades will include the integration of a high-resolution OPC to achieve higher size-resolved measurements of droplets. This improvement will provide more accurate droplet size binning, enabling better quantification of droplet growth and differentiation from interstitial particles of sizes larger than 200 nm.

# 565 Appendix A:

### A1 CFD simulations

The CFD simulations were conducted using the ANSYS software, Inc. products 2021 R2, with the Fluent module. The simulation process followed a standard workflow: geometry creation, meshing, boundary condition setup, running the simulation, and analyzing the results. A 3D design of the HCCNC plates and components was developed using the geometry component system. The mesh consisted of 737,604 total elements and 206,471 nodes, with an orthogonal quality. The minimum, maximum, average, and standard deviation of mesh sizes were 0.11, 1.0, 0.7, and 0.15 mm, respectively. To facilitate a smooth transition of meshing from the flat surface to the edges and back to the flat surface, five prism layers with a growth ratio of 1.2 were applied along the edges. The physics model comprises the 3D Reynolds-Averaged Navier-Stokes (RANS) equations, utilizing the realizable k- $\epsilon$  model along with the energy equation, steady-state conditions, and gravity effects. RANS equations are commonly used to compute time-dependent flows with external unsteadiness. The realizable k- $\epsilon$  model is an extension of the standard k- $\epsilon$  model and is an empirical framework based on transport equations for turbulence kinetic energy (k) and the specific dissipation rate ( $\epsilon$ ).

Figure A1, A2 and A3 presents the results obtained from the simulations, using the mesh and physics model described above. The boundary conditions applied included temperatures of 8°C for the top plate, 5°C for the bottom plate, and 20°C for the sheath and sample flow. The heat sink was set to the same temperature as the lower plate as it is screwed into the lower plate. The sheath flow was introduced at 1.3 L min<sup>-1</sup>, while the sample flow was set at 0.13 L min<sup>-1</sup>. The walls and injector material were set to AlSi10Mg and stainless steel, respectively. Convergence was achieved after approximately 60 iterations, with energy residuals below 10<sup>-6</sup> as the convergence criterion. Although there appears to be a slight disturbance in the supersaturation profile away from the centerline (Fig. A2), the turbulent kinetic energy is shown to be very low (Fig. A3), indicating that this is not due to turbulence. This disturbance likely arises from the limitations of the education version of ANSYS used in this study, which restricts the number of mesh grids, preventing higher-resolution simulations. A total of 17 different spacer and plate designs were simulated under the same boundary conditions, and the most optimal design—evaluated based on parameters such as laminar flow and temperature uniformity—was selected and is presented in this study, as discussed above.

**Figure A1:** Temperature simulation results of the HCCNC with the top (warm) plate placed at 8°C and the bottom (cold) plate at 5°C, while the sheath and sample flows enter the chamber at 20°C via the spacer and injector respectively. Temperatures exceeding 8°C have been clipped. (a) Displays the temperature gradient in the vertical mid-plane between the two plates. (b) Shows the temperature profile along the horizontal plane and at the middle of the chamber.

**Figure A2.** Supersaturation (SS) generated within the HCCNC when the top (warm) plate is at 8°C and the bottom (cold) plate at 5°C, while the sheath and sample flows enter the chamber at 20°C via the spacer and injector respectively. (a) Displays the supersaturation generated in the vertical mid-plane between the two plates. (b) Shows the supersaturation profile along the horizontal plane and at the middle of the chamber.

**Figure A3.** Turbulent Kinetic Energy (TKE) within the HCCNC when the top (warm) plate is at 8°C and the bottom (cold) plate at 5°C, while the sheath and sample flows enter the chamber at 20°C via the spacer and injector respectively. (a) Displays the TKE profile in the vertical mid-plane between the two plates. The sky-blue bordered inset presents a magnified view of a specific sky-blue region from the main figure. (b) Shows the TKE profile along the horizontal plane and at middle of the chamber.

### A2 Cross-section of chiller manifold

The cooling fluid is supplied to the plates through chillers connected via a chiller manifold (see Fig. 1). As shown in Fig. A4, the internal diameter of the manifold gradually decreases along the streamwise direction. This design compensates for pressure losses caused by friction over the length of the manifold. By reducing the diameter, the cross-sectional area decreases, causing the flow velocity to increase as per the continuity equation (Fox & McDonald, 2004) which helps maintain sufficient pressure to effectively transport the cooling fluid. This ensures an even distribution of cooling fluid across all four inlets of the plates.

Figure A4. Schematic illustrating the cross-section of the chiller manifold.

### A3 Intercomparison between OPC's at different flow rate

An OPC Model 804 (MetOne Instruments, Inc.) is used to detect droplets at the HCCNC outlet. While the OPC is calibrated for a flow rate of 2.83 L min<sup>-1</sup>, it was operated at 1.5 L min<sup>-1</sup> to ensure adequate residence time for

particles in the HCCNC, as the higher flow rate ( $2.83 \text{ L min}^{-1}$ ) reduces the time particle exposes to supersaturation conditions, affecting growth and detection. Intercomparisons between the HCCNC OPC and a reference OPC of the same model operated were conducted using room air. The results (Fig. A5) indicate that the HCCNC OPC measured a ~14% lower particle number concentration at the reduced flow rate (i.e. at 1.5 L min<sup>-1</sup>, blue data points). When both OPCs were operated at the same flow rate of  $2.83 \text{ L min}^{-1}$ , the average percentage difference was ~8% (red data points), which falls within the typical manufacturer-reported uncertainty of  $\pm 10\%$ . Consequently, to account for the reduced flow rate, the blue data points require a 15% adjustment to align with the reference OPC measurements which was also applied to our measurements (see Section 2.6).

**Figure A5.** Intercomparison of particle number concentrations measured by the HCCNC OPC and a reference OPC using room air at ambient temperature. The red data points represent measurements when both OPCs operated at 2.83 L min<sup>-1</sup> while the blue data points correspond to conditions where the HCCNC OPC operated at 1.5 L min<sup>-1</sup> and the reference OPC at 2.83 L min<sup>-1</sup>. The solid blue and red lines represent the fitted trend. The correction factor derived from the regression equation is given by  $OPC_{HCCNC} = 0.76*OPC_{ref} + 1.13$  when the HCCNC OPC was operated at 1.5 L min<sup>-1</sup>.

# A4 Particle activation, diffusional growth and sedimentation

Figure A6 illustrates the theoretical growth of a 97.1 nm particle following activation at a fixed temperature of 30  $^{\circ}$ C under various supersaturation (SS) conditions. The plot also depicts the particle residence time in the HCCNC relative to different injector positions (ix, x = 1-8). Figure A7 presents the activation, growth, and settling of a 97.1 nm ammonium sulfate particle as a function of residence time in a 0.15% supersaturation environment in the HCCNC.

**Figure A6.** Theoretical growth of activated ammonium sulphate particles with dry diameters of (a) 97.1 nm and (b) 191.4 nm, modelled as a function of supersaturation and residence time based on the growth equation from Rogers (1988) at 30 °C. The secondary x-axis (i1-8) represents the injector position at the HCCNC, corresponding to the particle residence time on the primary x-axis. Theoretical particle residence time is calculated by assuming a centerline particle velocity is 3/2 of the bulk velocity (parabolic velocity profile), with a total chamber flow rate of 1.5 L min<sup>-1</sup>, as the particle stream flows between two parallel plates. "i8-m" denotes the experimentally measured residence time at injector position 8.

**Figure A7.** Activation, growth, and settling of a 97.1 nm ammonium sulfate particle as a function of residence time in a supersaturation environment of 0.15% in HCCNC, modeled according to Rogers (1988). The grey strip represents the aerosol and droplet stream upon immediate activation, while the coloured strip corresponds to delayed activation. The primary y-axis denotes the vertical position of droplets between parallel plates, with colour indicating fall speed for the delayed activation case. The secondary y-axis represents droplet growth. The vertical black dashed line marks the measured residence time at injector position 8 in HCCNC.

Droplet growth following activation has been modeled based on Rogers (1988). Equations 5, 6, and 7 are employed to calculate the droplet growth.

$$r(t) = \sqrt{r_0^2 + 2\left(\frac{S-1}{F_k + F_d}\right)t}$$
 (5)

$$F_k = \frac{\rho_{water} L_v^{2^{\square}}}{K R_v T^2} \tag{6}$$

$$F_d = \frac{R_v T}{D_v e_{s,w}(T)} \tag{7}$$

 $F_k$  is the thermodynamic term representing latent heat release during condensation, while  $F_d$  refers to the vapor diffusion term associated with the transport of water vapor to the growing droplet. The initial dry particle radius,  $r_0^{\square}$ , is 48.5 nm, and the time step is denoted by t in seconds. S is showing the saturation ration, starting from 1. The temperature (T) is 30 °C, and the density of water ( $\rho_{\text{water}}$ ) is taken as 1004 kg m<sup>-3</sup>. The gravitational acceleration (g) is set to 9.81 m s<sup>-2</sup>. The viscosity of air ( $\eta_{\text{air}}$ ) is assumed to be 1.81 × 10<sup>-5</sup> Pa·s, while the density of air ( $\rho_{\text{air}}$ ) is taken as 1.1512 kg m<sup>-3</sup>. A dimensionless constant (T) is used, with a value of 1. The thermal conductivity of air (T) is specified as 0.02597 W m<sup>-1</sup> K<sup>-1</sup>, and the diffusivity of water vapor in air (T) is 2.75763 × 10<sup>-5</sup> m<sup>2</sup> s<sup>-1</sup>. The saturated vapor pressure of water (T) is 4246.81 Pa. The latent heat of vaporization of water (T) is taken as 2,575,000 J kg<sup>-1</sup>, and the gas constant for water vapor (T) is 461.5 J kg<sup>-1</sup> K<sup>-1</sup>.

#### A5 Relationship between supersaturation, temperature and maximum droplet number concentration

The maximum CCN count rate in the HCCNC varies as a function of supersaturation (SS%), temperature and particle concentration. When an aerosol particle activates into a cloud droplet, it rapidly grows in the supersaturated environment. This growth depletes the local water vapor field, which is continuously replenished by the SS source (wetted filter paper). However, competition for water vapor among growing droplets can set in if there are a high number of growing drops and affect the droplet growth rate. Figure A8 illustrates the critical droplet number concentration beyond which water vapor competition sets in a given supersaturation level. This calculation is based on theoretical considerations, where the available mass of water vapor is determined for a given supersaturation and temperature (per cm³). The total available water mass is then divided by the mass of an individual droplet to estimate the particle number concentration threshold. To ensure a realistic assumption, the droplet size after activation is considered to be 2.5 µm, as this represents the upper limit of the OPC size range used in this study. As shown in Figure A8, the particle number concentration threshold at which water vapor competition becomes relevant is lower at lower temperatures. This is attributed to the reduced absolute mass of water vapor available at colder conditions (approximately 8°C).

**Figure A8.** Maximum droplet number concentration as a function of supersaturation at different temperatures before water vapor competition becomes relevant. Circular markers represent the theoretically calculated data points, while the dashed line indicates the linear fit. The red, green, and blue markers correspond to supersaturation calculations at 30°C, 20°C, and 8°C, respectively.

- Code and data availability. No existing datasets have been used in this work. The data presented in the figures here will be
- made available through a permanent DOI through ETH library upon acceptance of the publication.
- Author contributions. MS designed components of the HCCNC and conducted the CFD simulations, with input from ZAK.
- MR machined spacer and injector of the HCCNC. MS and MR assembled the HCCNC. MR assembled the electronic interface
- and MS wrote a python code to control the electronics. MS conducted the validation experiments with input from ZAK. MS
- analysed the data and prepared the figures with input from ZAK. MS and ZAK interpreted the data. MS wrote the first draft
- of the manuscript with input from ZAK. ZAK conceived the idea of the low temperature HCCNC and MS designed the
- temperature control to achieve lower temperature and lower supersaturation. All authors reviewed the manuscript. ZAK
- supervised the project and obtained funding.
- Competing interests. The authors declare that Zamin A. Kanji is an associate editor for Atmospheric Measurement Techniques.
- Acknowledgements. MS and ZAK acknowledge funding from the SNSF grant number 197149. They also appreciate the helpful
- discussions with Nora Fahrenbach for her valuable contributions and feedback on the draft. MS acknowledges the support of
- Prof. Ulrike Lohmann, who allowed MS to work on the HCCNC and supported the project during delays. The authors also
- thank Athanasios Nenes for insightful discussions. Special thanks to Baptiste, Jie, Cuiqi, Fran, and Jorg for their initial
- introduction of the lab and instruments to MS.

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

- Limitations Affect Cloud Condensation Nuclei Activity Measurements Under Low Supersaturation, Geophysical
- Research Letters, 50, e2022GL101603, https://doi.org/10.1029/2022GL101603, 2023.

- Topping, D., Connolly, P., and Mcfiggans, G.: Cloud droplet number enhanced by co-condensation of organic
- vapours, Nat Geosci, 6, 443–446, https://doi.org/10.1038/ngeo1809, 2013.
- Topping, D. O. and McFiggans, G.: Tight coupling of particle size, number and composition in atmospheric cloud
- droplet activation, Atmos Chem Phys, 12, 3253–3260, https://doi.org/10.5194/acp-12-3253-2012, 2012.
- Vargaftik, N. B., Volkov, B. N., and Voljak, L. D.: International Tables of the Surface Tension of Water, J Phys
- Chem Ref Data, 12, 817–820, https://doi.org/10.1063/1.555688, 1983.
- Wexler, A. S. and Clegg, S. L.: Atmospheric aerosol models for systems including the ions H+, NH4+, Na+, so42-
- 798, NO 3-, Cl-, Br-, and H2O, Journal of Geophysical Research Atmospheres, 107,
- https://doi.org/10.1029/2001JD000451, 2002.

808

- Whitehead, J. D., Irwin, M., Allan, J. D., Good, N., and McFiggans, G.: A meta-analysis of particle water uptake
- reconciliation studies, Atmos Chem Phys, 14, 11833–11841, https://doi.org/10.5194/acp-14-11833-2014, 2014.
- Zelenyuk, A. and Imre, D.: On the effect of particle alignment in the DMA, Aerosol Science and Technology, 41,
- 112–124, https://doi.org/10.1080/02786820601118380, 2007.
- Zelenyuk, A., Cai, Y., and Imre, D.: From Agglomerates of Spheres to Irregularly Shaped Particles: Determination
- of Dynamic Shape Factors from Measurements of Mobility and Vacuum Aerodynamic Diameters, Aerosol
- Science and Technology, 40, 197–217, https://doi.org/10.1080/02786820500529406, 2006.