# Peer review of "Development of the Horizontal Cloud Condensation Nuclei 1"

_EGUsphere, 2025_

## Author Comment (AC1)

Reviewer comments have been reproduced in **bold** and author responses in regular typeface. Locations to modified text in the revised manuscript is presented in highlighted text.

**Review of**
**Development of a Horizontal Cloud Condensation Nuclei Counter (HCCNC) to detect particle activation at temperatures below 4°C and supersaturations below 0.05%**
**by**
**Mayur G. Sapkal, Michael Rösch and Zamin A. Kanji**

**This study presents the development of a CCN counter capable of measuring the CCN activity of particles at temperature as low as of 4°C and supersaturation (SS) levels down to 0.05%. The authors provided detailed design information and validated the instrument using ammonium sulfate particles. Overall, I find the work valuable and recommend it for publication after the following concerns are addressed:**

We sincerely thank the reviewer for their thoughtful and constructive review. We greatly appreciate the insightful questions, detailed suggestions and future guidance.

1. **While the newly developed HCCNC has been well validated using ammonium sulfate particles, how does it perform when measuring ambient particles? Adding experimental data or discussion regarding its application to real atmospheric aerosols would strengthen the work and make the study more comprehensive.**

   We appreciate the reviewer's valuable comment. Currently the HCCNC has not been used to evaluate ambient particle activation. To conduct field experiments is beyond the current scope of the manuscript and project. However, we did test the HCCNC with chamber experiments where mixed particles were used with ammonium sulphate and levoglucosan and address these in Q4 related to the reviewer concern. The sampling of the mixed particles can be understood as a proxy for ambient aerosol where organic and inorganic aerosol are internally mixed. In the next project phase of the HCCNC, the instrument will be deployed in the field. Given that the chamber worked well for the mixed particles, we do not foresee an issue in ambient operation as long as the particle concentration is below 7400 cm$^{-3}$ to avoid water vapour competition for activation and growth in the chamber (addressed in line 517-520 in the revised manuscript).

2. **The authors state that operating the CCNC at low SS allows for the activation of larger particles and provide an example where the critical SS for 111 nm ammonium sulfate particles is 0.13% (Lines 90–92). However, ambient URBAN aerosols are typically complex mixtures containing inorganics, organics, black carbon, dust, etc., and often exhibit lower hygroscopicity (kappa ≈ 0.3) compared to pure ammonium sulfate. This suggests that the D50 at SS = 0.13% for ambient particles would be significantly larger than 111 nm. Is it necessary to operate the CCNC at such a low SS? Would this low SS setting be more suitable for marine environments, where sea salt (e.g., sodium chloride) particles are much more hygroscopic? I recommend the authors clarify this point.**

   We agree with the reviewer that the lower supersaturations allowing the study of 200 nm particles is relevant for particles as hygroscopic as ammonium sulphate and would be relevant in marine environments. This also means for typically less hygroscopic aerosol such as in Urban environments particles even larger than 200 nm could be studied since the SS$_{crit}$ would be lower for larger particles compensated by their larger size. As recommended by the reviewer this point has been clarified in lines 511-514 of the revised manuscript.

3. **The authors state that operating the CCNC at low temperatures enables accounting for or capturing the co-condensation effect. However, co-condensation depends on the difference between particle composition activity and the saturation ratio of condensable gases, rather than temperature alone. Although lowering the temperature can decrease the saturated vapor**

**pressure of gaseous compounds, thereby increasing their saturation ratio and potentially enhancing co-condensation, this approach does not accurately reflect co-condensation processes under real atmospheric conditions. In fact, it may lead to an overestimation of the co-condensation effect compared to what occurs in the ambient environment.**

You are correct that co-condensation is complex and that operating at a single low temperature could overestimate the effect compared to ambient conditions. Especially if the HCCNC is operating temperature that are significantly lower than ambient cloud temperatures. This could lead to an overestimation of CCN counts.

However, the key feature of the HCCNC is not its ability to generate the supersaturation at one low temperature (let's say 4°C) only, but its wide, controllable temperature range (4°C to 35°C) for supersaturation generation. This unique flexibility allows researchers to either match specific ambient thermal conditions or to systematically isolate and study the influence of temperature on aerosol activation and co-condensation.

4. **The maximum AF in Figure 4 ranges between approximately 0.7 and 0.9, which the authors attribute to uncertainties between CPC and OPC measurements. If this is the case, one would expect similar maximum AF values under different experimental conditions. Why does the maximum AF vary between 0.7 and 0.9? I am concerned that the AF could be even worse when measuring complex ambient particles. Could the authors provide some ambient particle measurement data to illustrate the instrument's performance in real-world conditions?**

This is a valid question. The differences in the AF could arise from the differences in counting efficiency between the CPC and OPC which has been shown to be uncertain in previous studies as well (Kumar et al., 2003). To demonstrate the instrument's capability with ambient-type aerosols, we conducted experiments with mixed ammonium sulfate and levoglucosan particles. The organic fraction from the Aerosol Mass Spectrometer (AMS) and the activated fraction (AF) measured by HCCNC over time are shown in plots below. These results indicate that for mixed inorganic-organic particles, similar to ambient aerosols, the AF reaches about 0.9. This work is in preparation for a follow up study to be submitted soon.

[Figure]

5. **The current title implies that the HCCNC can measure at temperatures BELOW 4°C and SS lower than 0.05%. However, based on the manuscript, the system achieves measurements at 4°C and 0.05% SS, not below these thresholds. Please revise the title and corresponding statements in the abstract to reflect the actual capabilities of the instrument.**

Agree. We have now updated the title accordingly.

References:

Pradeep Kumar, P., Broekhuizen, K., and Abbatt, J. P. D.: Organic acids as cloud condensation nuclei: Laboratory studies of highly soluble and insoluble species, Atmos. Chem. Phys., 3, 509–520, https://doi.org/10.5194/acp-3-509-2003, 2003.

---

## Author Comment (AC2)

Reviewer comments have been reproduced in **bold** and author responses in regular typeface. Locations to modified text in the revised manuscript is presented in highlighted text.

**Review of**
**Development of a Horizontal Cloud Condensation Nuclei Counter (HCCNC) to detect particle activation at temperatures below 4°C and supersaturations below 0.05%**
**by**
**Mayur G. Sapkal, Michael Rösch and Zamin A. Kanji**

**This paper introduces a horizontal CCN counter (HCCNC) designed for CCN measurements under low temperature and low supersaturation conditions. The authors provide detailed descriptions of the instrument's construction, experimental setup, validation, and associated uncertainties. The device is expected to improve the accuracy of CCN measurements based on its newly designed compact and lightweight chamber. However, due to the challenges of measuring CCN at low supersaturation, the technical evidence provided is currently insufficient to fully demonstrate the instrument's performance under these conditions. The manuscript falls well within the scope of AMT and I recommend it for publication after the following comments are addressed.**

We sincerely thank the reviewer for their thoughtful and constructive review. We greatly appreciate the keen observations, insightful questions, and detailed suggestions, as well as the references provided. These comments have helped us improve the clarity and scientific rigor of the manuscript.

**1) The improvement in residence time ($\tau$) offered by the HCCNC appears to be limited. The residence time depends on the cloud chamber's volume (V) and the airflow rate (Q), roughly following the relationship $\tau \propto V/Q$. Commercial CCNC use cylindrical chambers (about 500 mm long, 22.7 mm in diameter), with a volume of around 0.2 L and a flow rate of about 0.5 L/min. In contrast, the HCCNC uses a new designed chamber (410 mm long, 210 mm wide, 13 mm low) with a volume of about 1.1 L and a flow rate of 1.5 L/min. While the HCCNC has roughly 5.5 times the volume and 3 times the flow rate of typical CCNCs, its estimated residence time is only about 80% longer—not even twice as long.**

We thank the reviewer for highlighting this point and would like to clarify that the particle residence time in HCCNC is measured via a pulse test (following the method of Brunner and Kanji, 2021; Garimella, S. et al., 2017) which provides a more accurate assessment of the residence time. At a flow rate of 1.5 L/min and injector position 8, the HCCNC exhibited a residence time of 21 seconds. In comparison, DMT CCNCs typically have residence times ranging from 6 to 12 seconds depending on the flow rate (Rose et al., 2008), making the HCCNC's residence time approximately 350% to 175% longer. Furthermore, the successful activation of 200 nm ammonium sulfate particles at 0.05% supersaturation (Figure 4) demonstrates that the current residence time is sufficient for capturing activation behavior under such low supersaturation conditions. In future work there is potential to further increase residence time and enable operation at even lower supersaturations. We have now added this point in the revised manuscript line number: 557-559.

**Since the time required for droplet activation increases rapidly as SS decreases, the HCCNC still needs to rely on droplet size distribution measurements to identify CCN under low SS conditions, as discussed in Section 3.1.2. However, the current study does not fully demonstrate how well the HCCNC performs in identifying CCN at low SS down to 0.05%. Figure A6 shows how the device uses the calculated critical droplet size to distinguish CCN. But the lowest SS tested and verified is only 0.1%.**

Time required for activation should not be related to the lower SS, however the growth of the droplet to our detecting size bin is dependent on the SS and residence time. We believe this

comment is a misunderstanding regarding the purpose of Figure A6. Figure A6 is not used to determine critical supersaturation or droplet size thresholds. Rather, it presents a theoretical droplet growth trajectory of an ammonium sulfate particle—regardless of size—after activation, either in a cloud environment (or within the HCCNC), under a given supersaturation. This is based on established principles of mass transfer (see section A4) and is intended only to conceptually illustrate the growth dynamics of an activated droplet. The actual critical supersaturation levels, including those as low as 0.05%, are determined independently using the measured temperature profiles inside the chamber. We did, in fact, successfully verify activation at this low supersaturation as shown in Figure 4. It appears that the original intent of Figure A6 may not have been clearly conveyed. To clarify this, we have revised the figure, its caption and related text (revised manuscript line number: 330-332, 635, 641-646). Furthermore, we have updated the figure itself to include the 0.05% SS level and have also added the theoretical growth of the activated particles of 97.1 and 191.4 nm sizes, as this addition aligns with the measurements sizes and supersaturation at which operated the HCCNC during the validation.

**2) Another concern is the OPC used in the HCCNC. It only has four size bins (>0.5, >0.7, >1.0, and >2.5 µm), and it's unclear whether this limited resolution is enough to accurately capture the droplet size distribution. This is especially important at low SS, where critical droplet size may be varied or the growth difference between activated and non-activated particles may be subtle. In such cases, it's uncertain whether the device can reliably tell CCN apart from interstitial aerosol based on critical droplet size.**

We agree that using only four size bins (>0.5, >0.7, >1.0, and >2.5 µm) might limit detailed characterization of droplet size. However, for larger particles—such as 200 nm ammonium sulfate at 0.05% supersaturation—a clear activation transition was observed in the >2.5 µm bin (Figure 4). This allows effective separation between interstitial and activated aerosol within the current OPC resolution, and is further supported by EAIM modeling, which confirms that hygroscopic growth alone cannot explain such size increases. As such the current resolution is demonstrated to be enough to capture CCN activation even of large particles growing at low supersaturations since they can be detected in the 2.5 µm channel and result in accurate $SS_{crit}$. Additionally, the OPC settings can be modified to set the largest size channel to 5 µm. We are limited in detecting sizes between > 1 µm and > 2.5 µm. We have acknowledged that a different OPC can be deployed with the HCCNC that uses different flow rates or OPC size binning and resolution (line 534 – 535 and 562-564 in revised manuscript).

**3) Finally, while the paper discusses SS uncertainty at a high SS value (0.203%, Fig. 2a), it does not clearly evaluate or report SS accuracy or uncertainty in the lower SS range (SS < 0.1%), which is critical. Even small absolute errors in SS can cause large differences in the fraction of particles that activate at low SS, so precise control and measurement of SS is very important in this range.**

We would like to clarify that SS uncertainty was indeed calculated for both high and low supersaturation conditions using the same methodology, as applied for SS = 0.203% in Figure 2a. The error bars owing to such uncertainty calculated in Fig. 2 are shown in Figure 4c, where we present the activation curves for the lowest temperatures and the uncertainties are the highest. The uncertainty bars do not show up at the higher temperatures (20–30 °C, Figs. 4a and 4b) because the derived uncertainty is very small compared to the graphical resolution. We have now referred to the larger uncertainty at the lower supersaturation in the caption of Fig. 2 and the caption of Fig. 4. In addition, we add the uncertainty in $SS_{crit}$ to Fig. 4 to demonstrate the range across different SS.

**3) The title and the abstract: I think it's not "below 4°C" and "below 0.05%", but "down to 4°C" and "down to 0.05%".**

We fully agree with the reviewer. Adding the phrase "down to" twice reads a little clunky, so we have updated the title to "Development of the Horizontal Cloud Condensation Nuclei Counter (HCCNC) to detect particle activation down to 4°C temperature and 0.05% supersaturation".

**4) L14: Please give a reference about statement of "streamwise CCNC struggle to achieve supersaturations below 0.13%".**

We have provided the relevant reference and example in the main text of the revised manuscript (Revised manuscript line number: 89) since the abstract cannot contain references.

**5) L36: Please specify "a considerable degree" -.**

We have clarified the statement by specifying that 'a considerable degree' refers to measured critical supersaturations being in agreement with theoretical predictions based on Köhler theory such that closure is achieved between measured and modelled CCN concentrations. We have clarified this point in lines 129-131 of the revised manuscript.

**6) L85–89: I have question on the statement that "the residence time in the streamwise CCNC is fixed for a given flow rate, making operation below 0.13% supersaturation impractical." In fact, the residence time can be increased by reducing the flow rate (Lance et al., 2006).**

We agree that the residence time in the DMT-CCNC can indeed be increased by reducing the flow rate. However, even at the lowest practical flow rates, the maximum achievable residence time in the DMT-CCNC is approximately 12 seconds (Rose et al., 2008). Under low supersaturation conditions (e.g., <0.13%), this residence time is often not sufficient to grow activated droplets to sizes large enough to be reliably distinguished from unactivated, interstitial particles (Tao et al., 2023). This limitation arises because at low SS, droplet growth after activation is slow, as the supersaturation is the sole driving force for growth. For this reason, the operational lower limit of ~0.13% SS, as specified in the DMT-CCNC manual, reflects not just flow-rate constraints but also the practical difficulty of resolving activated droplets under such conditions.

**7) I also question the statement that growth kinetics due to high particle concentrations limit the streamwise CCNC's ability to study atmospherically relevant particle sizes and chemical compositions. When the CCNC is placed downstream of a DMA—as in this study and in many former CCN studies—the particle concentration entering the CCNC can be significantly reduced. This setup helps minimize growth kinetics limitations.**

While upstream dilution or particle classification (e.g., with a DMA) can reduce particle concentrations entering the CCNC, if these concentrations still remain above approximately 6000 cm$^{-3}$, growth kinetics can limit the instrument's ability to accurately measure activation (Fig. 51, DMT CCNC manual (DOC-0086 Revision I-2, pg. 107)) at atmospherically relevant supersaturations (typically below 0.1%). Additionally, this reliable lower limit for supersaturation measurement increases to about 0.2% in some cases, potentially obscuring the activation of larger atmospheric particles.

**8) L114–115: As reported by Tao et al. (2023), CCN-active droplets can still be distinguished from interstitial aerosols by calculating their growth at supersaturations below 0.15%, even when the residence time in the CCNC is not long enough for full activation.**

Thank you for the valuable feedback and the reference. We agree that the method described by Tao et al. (2023) provides a way to computationally correct for kinetic limitations.

However, we think the primary drawback of this calculation-based approach is its high sensitivity to parameters that are unknown and highly variable for ambient aerosols.

Specifically, the calculation requires assuming a value for the water vapor accommodation coefficient ($\alpha$), which, can vary significantly with chemical composition, particularly due to organic compounds. For complex ambient aerosols, assuming a correct $\alpha$ can be a major source of uncertainty that our direct measurement approach avoids. Another source of uncertainty is assuming that activation occurs instantaneously upon aerosol entry to the chamber and that the entire residence time is available for growth, which may not be true and overlooks the time needed for the aerosol to equilibrate to the chamber inner conditions. As such to be able to measure the droplets at the low SS is direct evidence of the $SS_{crit}$ of particles activating at low SS.

**9) L126-128: This sentence is not clear enough.**
We agree and have revised the sentence in the revised manuscript (lines 129-131).

**10) L156–160: Buoyancy-driven air movement becomes significant only when the temperature difference is greater than 10 K (Rogers, 1988; Stetzer et al., 2008), which corresponds to high supersaturation conditions (SS > 0.4%) in the streamwise CCNC. At lower SS levels, the effect of buoyancy-induced air movement in the streamwise CCNC can be considered negligible.**
We thank the reviewer for this clarification. We have updated the manuscript to indicate that buoyancy-driven air movement becomes significant when the temperature difference exceeds 10 K in the streamwise CCN column (Revised manuscript line number: 161-162).

**11) L326: Please give more details about the diffusional growth calculations in Rogers (1988).**
The diffusional growth calculations are described in detail in Section A4; however, we inadvertently omitted this reference in the main text. We thank the reviewer for catching that! We have now added this cross-reference to the manuscript for clarity (Revised manuscript line number: 330-331).

**12) L344: This delay may be stronger at lower SS. How would this affect the measurement of HCCNC?**
We appreciate the reviewer's observation regarding the potential for slower vapor and thermal equilibration at lower supersaturations. We agree that at lower SS this delay could be longer, however we have characterised the chamber for SS of 0.05% with 191.4 nm ammonium sulphate particles and can detect the activated droplets in the 2.5 μm channel. The delay implies that the droplets do not grow as large as theory would predict if we assumed the entire residence time is available for growth. This is to our advantage because it means the droplets would not settle out of the flow through sedimentation and would be detected. This is the exact reason why the experimental validation is combined with the theoretical calculations in Appendix A4.

**13) L422–424: It is unclear why a counting uncertainty of ±10% for both the CPC and OPC results in a reported AF uncertainty of 14%. In my view, a total uncertainty within ±20% is reasonable. I suggest revising the sentence as follows: Given that both the OPC and CPC used in the validation experiments have counting uncertainties of ±10%, the combined relative uncertainty in AF should be within ±21%, and thus the reported ±14% uncertainty is reasonable.**
The reported uncertainty of ±14% comes from the fact that the AF is calculated by dividing the OPC droplet count by the CPC aerosol count. As such the uncertainties in these parameters need to be propagated into the AF. To do this, the relative error needs to be propagated. Since both the CPC and OPC have counting uncertainties of ±10%, The relative error in the AF based on standard error propagation for independent measurements is calculated using the root-sum-square method:

$$\sqrt{(10^2+10^2)} = 14.2\%$$

This approach follows standard error propagation rules for independent uncertainties and supports the reported ±14% value which has been rounded up to 15% (to be conservative) in the revised manuscript. This addition now can be found in the updated manuscript at line 432-434.

**14) L478–481: Both CCN activation and hygroscopic growth of ammonium sulfate reflect its hygroscopicity, but under different levels of water vapor saturation. A recent study using a low-temperature hygroscopicity tandem differential mobility analyzer (Low-T HTDMA) measured the hygroscopic growth of ammonium sulfate under low temperatures (Cheng and Kuwata, 2023). I suggest discussing how these results compare with the findings in this study.**

We thank the reviewer for the suggestion. Following this recommendation, we have incorporated a comparison. Our findings show that the kappa ($\kappa$) value derived from HCCNC measurements at 8 °C is 0.42±0.37. This result is comparable to the $\kappa$ of 0.49 measured at 10 °C by Cheng and Kuwata (2023) using a Low-T HTDMA. To provide additional context, our HCCNC-derived $\kappa$ at 20 °C is 0.47±0.03, which shows agreement with the value of 0.46±0.01 reported by Gysel et al. (2002) at the same temperature. The larger uncertainty in our kappa measurement at 8 °C is attributed to the associated uncertainty in supersaturation at lower temperatures. This addition now can be found in the updated manuscript at line 493-498.

**15) Figure 5: The effects of non-ideal behavior of ammonium sulfate on CCN activation and related measurements have been investigated by Rose et al. (2008). I recommend using the parameterization of the Van't Hoff factor based on solute molality, as described by Young and Warren (1992) and Frank et al. (2007) mentioned in Rose et al. (2008).**

We thank the reviewer for the insightful suggestion and for pointing us to the relevant references. Following the recommendation, we incorporated the parameterization of the van't Hoff factor based on solute molality, specifically using Equation A25 from Rose et al. (2008), derived based on Frank et al. (2007). Applying this formulation, we obtained a Van't Hoff factor of ~1.94 for ammonium sulfate under our experimental conditions. This adjustment further improved the consistency of our results with theoretical Köhler predictions, enhancing the robustness of our analysis. A new plot and associate description reflecting these updates has been included in the revised manuscript. (Figure: 5, Line Number: 477-478, 487-488)

**16) Figures A1 and A2: It is not clear why the spatial distribution of temperature and supersaturation downstream of the injector appears asymmetric after aerosol injection. Could this be due to a pressure drop along the aerosol flow path inside the injector in the direction of the main airflow?**

The asymmetry in temperature and supersaturation distributions downstream of the injector is very likely due to a pressure gradient along the aerosol flow path inside the injector. Since the injector is closed at one end and open at the other (at the sample inlet), the internal pressure is higher at the closed end and lower at the open end. This creates non-uniform ejection velocities along the slit, with slightly higher sample flow exiting near the high-pressure (closed) end. The resulting asymmetric particle flow distribution leads to localized variations in temperature and supersaturation fields.

**17) Figure A5: Why is the OPC count lower at lower flow rates? Could this be due to coincidence errors?**

It is possible that coincidence is the reason since operating the OPC at lower flowrates would imply that the particles are spending more time in the laser detection volume than at higher flow rates So more particles could be entering the sample volume before detected particles have fully exited the sampling volume. We have corrected for the difference in concentrations by performing a systematic comparison

by operating at the two different flow rates, achieving a correction factor of 10% (line 621-625 in the revised manuscript).

References:

Brunner, C. and Kanji, Z. A.: Continuous online monitoring of ice-nucleating particles: development of the automated Horizontal Ice Nucleation Chamber (HINC-Auto), Atmos. Meas. Tech., 14, 269–293, https://doi.org/10.5194/amt-14-269-2021, 2021.

Garimella, S., Rothenberg, D. A., Wolf, M. J., David, R. O., Kanji, Z. A., Wang, C., Rösch, M., and Cziczo, D. J.: Uncertainty in counting ice nucleating particles with continuous flow diffusion chambers, Atmos. Chem. Phys., 17, 10855–10864, https://doi.org/10.5194/acp-17-10855-2017, 2017.